# Is Radiographic Aftercare Obsolete? How Testing Positive for ctDNA Can Be a Precedent for Late Relapse, Even in Low-Risk Hormone-Receptor-Positive Breast Cancer

**DOI:** 10.3390/ijms26178498

**Published:** 2025-09-01

**Authors:** Kerstin Pfister, Sophia Huesmann, Angelina Fink, Thomas W. P. Friedl, Franziska Mergel, Henning P. Schäffler, Andreas Hartkopf, Stefan Lukac, Kristina Veselinovic, Forca Mehmeti, Nathan Campbell, Christodoulos Pipinikas, Katharina Deininger, Ambros Beer, Stefanie Lorenz, Meinrad Beer, Lisa Wiesmüller, Wolfgang Janni, Sabine Heublein, Brigitte Rack

**Affiliations:** 1Department of Gynecology and Obstetrics, University Hospital Ulm, 89075 Ulm, Germany; 2Department of Gynecology and Obstetrics, SLK-Kliniken Heilbronn, 74078 Heilbronn, Germany; 3Department of Women’s Health, Tübingen University, 72076 Tübingen, Germany; 4NeoGenomics Babraham Research Campus, Cambridge CB22 3AT, UK; 5NeoGenomics Research Triangle Park, Durham, NC 27713, USA; 6Department of Nuclear Medicine, University Hospital Ulm, 89081 Ulm, Germany; 7Department of Radiology, University Hospital Ulm, 89081 Ulm, Germany

**Keywords:** breast cancer, liquid biopsy, ctDNA, molecular relapse, MRD

## Abstract

Current aftercare after early breast cancer overlooks recent evidence on circulating free tumor DNA (ctDNA). In this case report, we present a patient with low-risk hormone-receptor-positive breast cancer. ctDNA was first detected using a tumor-informed approach 12 months after the initial diagnosis and remained positive throughout adjuvant therapy with letrozole, while routine staging examinations showed no signs of relapse. Clinical relapse was ultimately identified nearly six years after the initial diagnosis, resulting in a lead time of four years and nine months. Current studies on ctDNA in the adjuvant setting have been conducted in high-risk cohorts and have shown a median molecular lead time of 8.9–12.4 months. Our study supports the need for large randomized clinical trials involving low-risk breast cancer patients to drive changes in clinical practice.

## 1. Introduction

Due to significant improvements in the early detection and treatment of breast cancer, early breast cancer (eBC) is now frequently curable, with a 10-year survival rate through all subtypes of 83% [1], highlighting the importance of follow-up care. Current clinical practice for patients with early breast cancer consists of clinical assessments, combined with regular mamma-sonography and mammography. Distant relapses are detected in most cases after the manifestation of metastasis-related symptoms. Evidence for this guideline includes two large studies carried out in the 1990s, indicating that intensified surveillance (via radiographic or serological examinations) does not improve overall survival [2,3].

The identification of new biomarkers, such as circulating tumor cells (CTC) and cell-free tumor DNA (ctDNA) that can be accurately and reproducibly detected in blood, has gained significant scientific interest in oncology in the last few years. Cell-free tumor DNA is genetic material that has been shed from tumor cells into blood circulation, due to necrosis, apoptosis and active release [4,5]. Whereas only a small proportion of total cell-free DNA consists of tumor DNA, small amounts can be detected in the context of molecular residual disease (MRD) using highly sensitive assays.

While several studies have provided evidence for the prognostic role of ctDNA in detecting molecular relapse ahead of clinical confirmation, the majority of these are focused on high-risk cohorts of eBC [6,7,8]. Low-risk eBC is commonly cured and relapses only occur occasionally, mostly with a latency of several years. In this case report, we show that locoregional relapse in a low-risk hormone-receptor-positive early breast cancer patient is preceded by ctDNA positivity by almost five years.

## 2. Results

In 2017, a 61-year-old female patient presented with a Luminal B lobular-invasive carcinoma G2 of the breast (pT1c, pN1a (1/2 sn), M0, R0, L0, V0, Pn0), ER (estrogen receptor) 100%, PgR (progesterone receptor) 0%, HER2 0, Ki-67 10%. The patient received breast-conserving surgery with the removal of the sentinel lymph node. An Endopredict test was performed and revealed an EPclin Score of 3.31, thus allocating the patient to the low-risk cohort, and no adjuvant chemotherapy was administered. Adjuvant radiation lasted until four months f/u (follow-up). In parallel, the patient started endocrine therapy with an aromatase inhibitor (AI, letrozole), alongside osteoprotective therapy with a bisphosphonate (zoledronic acid).

ctDNA was detected in the first retrospectively tested sample, taken one year after the primary diagnosis (1 year f/u) and throughout all subsequent collection timepoints (see Figure 1). The patient showed no clinical signs of relapse and oncological follow-up was performed in accordance with German national guidelines consisting of regular ultrasounds of both breast and lymph nodes and mammography, without any findings.

From four years and one month f/u, upon knowledge of the ctDNA positivity, staging examinations with F18-FDG (2-[18F]fluoro-2-deoxy-D-glucose)-PET/CT scans or CT thorax/abdomen and bone scans were performed every six months (see Figure 1). Additionally, prospective plasma ctDNA testing was performed (5 y 3 m (months) f/u and 5 y 9 m f/u). The tests showed positivity for ctDNA and increasing eVAF (estimated variant allele frequency) values throughout, whilst imaging showed no signs of local or locoregional relapse (see Appendix A).

At 5 y 9 m f/u and under continuous therapy with letrozole, F18-FDG-PET/CT examination indicated suspicious, infraclavicular and retropectoral lymph nodes (SUVmax (standardized uptake value maximum) 3.3) that were assessed as BI-RADS (Breast Imaging Reporting and Data System) 4C, following ultrasound. Histological examinations revealed a late relapse of the lobular-invasive carcinoma, ER 100%, PgR 0%, HER2 0, Ki-67 < 10%, Luminal B. The lead time from positive ctDNA to clinical relapse was consequently 4 years 8.9 months.

After careful discussion with the patient, she decided against a second surgical approach and opted for systemic therapy instead. Therefore, a combined endocrine therapy of fulvestrant plus CDK 4/6 inhibitor (abemaciclib) was initiated. The following ctDNA test (6 y f/u) revealed a ctDNA decline below the detection level (see Figure 1A), and staging examinations have shown stable disease for nine months (see Figure 1C).

The analysis of somatic variants (Figure 2) revealed an increase at 5 y 3 m f/u in the percentage of variant allele frequency (%VAF) of two variants in the TTN-AS1 and IGSF1 genes, immediately preceding the detection of positive findings in the PET-CT scans conducted at 5 y 9 m f/u (Figure 1A(III)). These genes have previously been shown to be involved early in the initiation of metastasis [9,10]. At 5 y 9 m f/u, the highest eVAF values were measured simultaneously to the first radiological signs of relapse (see Figure 1B(III)), while the detection of TTN-AS1 and IGSF1 declined. Thereafter, ctDNA levels cleared whilst CT scans showed a stable disease. Due to toxicity from the oncological treatment (interstitial lung disease caused by abemaciclib), the patient discontinued treatment with abemaciclib at seven years after initial diagnosis and has since been receiving endocrine monotherapy with fulvestrant. Regular CT scans show stable disease.

## 3. Discussion

Current aftercare in breast cancer neglects the advantages of liquid biomarkers. The value of ctDNA in detecting molecular relapse ahead of clinical evidence in the adjuvant setting has been shown in a variety of mainly retrospective studies. There are only a few prospective studies with sensitivity rates of 89–100%, specificity rates of of 97–100%, and a median molecular lead time of 8.9–12.4 months [6,7,8].

However, these studies were conducted within high-risk populations. In the studies by Coombes et al. [6], Lipsyc-Sharf et al. [8], and Garcia-Murillas et al. [7], patients had received neoadjuvant/adjuvant chemotherapy. Lipsyc-Sharf et al. [8] included only high-risk HR-positive, HER2-negative patients, with high-risk being defined as T3/4, N2/3; T1/2 only when there are at least three lymph nodes involved. A meta-analysis has underscored the high value of ctDNA in the after-care setting, with a hazard ratio of 14.04, 95% CI 7.55–26.11, in univariate analysis [13]. Small tumors (<T2) and marginal lymph node involvement (<3 positive lymph nodes), however, were not commonly part of these trials.

In this case report, whilst acknowledging the well-known limitations associated with single-patient observations and individual variability, we provide the first evidence for the clinical value of monitoring ctDNA in the detection of disease relapse in the follow-up care setting in low-risk breast cancers. In analogy to the inclusion criteria of the NATALEE study [12], this patient might have become eligible for a treatment with ribociclib as adjuvant therapy in current practice, regardless of a liquid biopsy. Importantly, however, longitudinal plasma ctDNA monitoring has the potential to identify early molecular relapse ahead of clinical evidence, thus enabling timely intervention before clinical symptoms become overt. By showing positive eVAF values years before relapse, this patient could have benefited from intensified aftercare. Upon the 29-fold eVAF increase in between 4 y 1 m f/u and 5 y 9 m f/u (see Appendix A), intensified adjuvant therapy could have been initiated at the period of maximum necessity. Even if case reports based on clinical history and outcomes of one individual patient have to be interpreted with care, the data reported here highlight the potential and importance of long-time surveillance following the diagnosis of early low-risk breast cancer.

Individualized, intensified surveillance following primary therapy in early breast cancer and post-adjuvant treatment at the time of greatest need could reduce the rate of clinical relapse and thus improve overall survival. Large prospective trials are needed to assess liquid-biopsy-based aftercare on a population-based approach. One example is the SURVIVE study (NCT05658172), a randomized surveillance study. Having started in 2022, it aims to compare guideline-based standard surveillance vs. liquid-biopsy-based intensified surveillance in 3500 women with intermediate to high-risk early breast cancer [14]. Economic considerations, including the number needed to treat (NNT), will be critical in evaluating the feasibility and utility of such strategies.

## 4. Materials and Methods

The patient in this case report was part of a feasibility pilot study for the SURVIVE-Study (*Standard Surveillance versus Intensive Surveillance in Early Breast Cancer*, NCT05658172). As part of the study, tumor material, as well as plasma, was collected at 4 timepoints (12/36/48/62 months after primary diagnosis) and plasma was analyzed using RaDaR, a highly sensitive, tumor-informed liquid biopsy assay.

RaDaR panel design

A formalin-fixed, paraffin-embedded (FFPE) tumor sample from the patient’s resection specimen was submitted to NeoGenomics Inc. (Durham, NC, USA) for whole-exome sequencing (WES). Tumor area and cellularity (20–80%) were assessed on hematoxylin and eosin (H&E)-stained slides. Tumor DNA was extracted from approximately 10 sections (10 μm thickness) using the Maxwell RSC DNA FFPE Kit (Promega, Madison, WI, USA) and quantified via a Quanti-IT dsDNA BR assay (Invitrogen, Carlsbad, CA, USA). WES was performed as previously described [8,15] using the KAPA Hyper Prep Kit (KAPA Biosystems, Wilmington, MA, USA) and unique dual-indexed (UDI) adapters for library preparation. Exome enrichment was performed with the IDT xGen Exome Research Panel v1.0, and the enriched library was sequenced on the Illumina HiSeq4000 platform (San Diego, CA, USA).

Data processing, performed as previously described [16], included alignment to the human genome (hg38), duplicate marking, and somatic variant calling using a proprietary pipeline; germline variants were filtered using public SNP databases. A personalized RaDaR^®^ assay was designed for this patient using 53 prioritized tumor-specific, WES-derived variants. The variants were prioritized and selected based on the read depth and allele frequency of each variant to assemble a variant set best suited for multiplex PCR and ensure the optimal amplification of target regions to allow for the sensitivity detection of plasma ctDNA. Full details of the entire RaDaR workflow from WES to plasma calling have been previously published [8,15,16]

Panels also included a panel of common, population-specific SNP controls (Integrated DNA Technologies, Coralville, IA, USA). Panel performance was validated via the NGS of tumor DNA, reference standards, and negative controls using the Illumina iSeq100.

b.Plasma ctDNA test

The panel was used to assess the presence of ctDNA in longitudinally collected samples taken between 2018 and 2023. Plasma samples were analyzed retrospectively (samples of 2018 and 2019) and, upon findings of positive ctDNA, prospectively (from 2020 onwards).

Whole-blood samples were collected in 4 × 10 mL PAXgene^®^ blood ccfDNA CE-IVD tubes (BD, Heidelberg, Germany) and processed to plasma within a median of 2 days (range: <1–7 days) from collection, as previously described [17]. Briefly, blood samples were processed with an initial centrifugation step at 2000× *g* for 10 min at 20 °C. The resulting plasma supernatant was then subjected to a second centrifugation (same settings) to remove any cellular debris and stored at −80 °C. Control genomic DNA was isolated from the cell pellets of the first centrifugation step after Ficoll Paque TM Plus (GE Healthcare, Chicago, IL, USA) gradient centrifugation using a standard protocol [18,19].

Plasma circulating cell-free DNA (cfDNA) was extracted at NeoGenomics Laboratories, Inc. (Durham, NC, USA) using a magnetic-bead-based protocol on the Hamilton Microlab STAR platform. No-template controls were included in the extraction of each batch to monitor for contamination. cfDNA and control genomic DNA were quantified using droplet digital PCR (ddPCR; Bio-Rad QX200, Hercules, CA, USA).

Following panel quality control, plasma cfDNA was amplified via multiplex PCR on the qualified personalized panel. Matched control genomic DNA, as well as negative and positive amplification controls, was included to exclude germline variants, clonal hematopoiesis-related mutations, and technical artifacts.

The amplicon products were barcoded and resulting libraries were pooled and combined with PhiX prior to ultra-deep sequencing on the Illumina NovaSeq 6000 platform.

Raw data were converted to FASTQ and reads were aligned to the human genome (hg38) using the bwa-mem software package (version 0.7.12-r1039) and analyzed using proprietary software to count mutant and reference alleles.

Plasma ctDNA calling was based on variants that remained after excluding those that failed validation in the patient’s tumor DNA during panel qualification, as well as those identified in the matched control genomic DNA, thereby minimizing false positive results. Moreover, precise knowledge of the number of remaining usable variants enabled an accurate determination of each sample’s limit of detection. The process of calling a sample positive or negative for residual disease has been previously described [15,16,20]. Briefly, a statistical framework was applied to evaluate the background noise for each remaining variant and to assess the statistical significance of the observed mutant counts. The contributions of these individual variants are then integrated across the entire set of validated variants. A sample is classified as positive for residual disease if the cumulative statistical score from all validated variants exceeds a predefined threshold established during the analytical validation of the RaDaR assay. Tumor fraction was then reported as the % estimated variant allele frequency (%eVAF), with a demonstrated assay sensitivity of 95% at a median VAF of 0.001%.

## Figures and Tables

**Figure 1 ijms-26-08498-f001:**
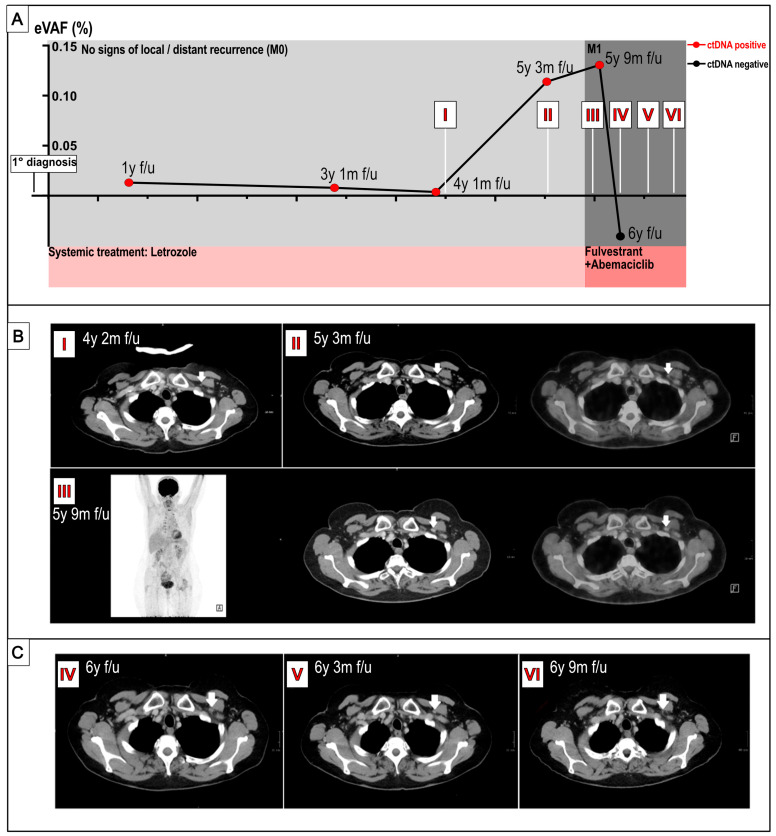
Time course plot (**A**) of ctDNA % eVAF levels and concurrent imaging ((**B**) during MRD, (**C**) after clinical recurrence) over time. (**A**) Time points of ctDNA evaluation are shown as dots, and dates of imaging procedures are depicted as Roman numerals (**I**–**VI**). (**B**,**C**) Computer tomography scans as well as F18-FDG-PET/CT (left CT, right fusion of CT and PET, (**B**)). Retropectoral lymph nodes, indicated by arrows, show no increased FDG uptake (**B**). CT scans show stable disease upon treatment with abemaciclib and fulvestrant (**C**). Locus of relapse again indicated with white arrows. (**C**).

**Figure 2 ijms-26-08498-f002:**
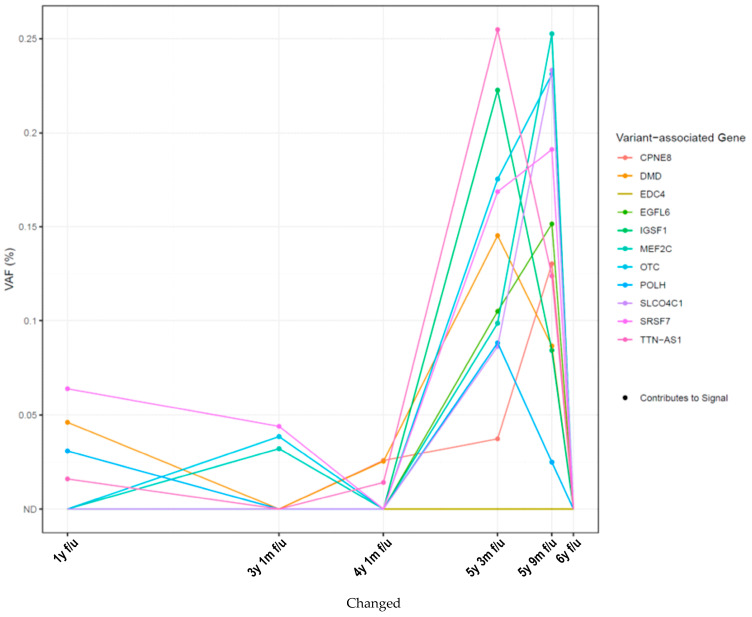
Longitudinal monitoring of variants contributing to positive ctDNA signal. WES-detected variants were used in the personalized RaDaR assay development for monitoring plasma ctDNA status over time (as previously described [11,12]). The figure shows the genes with which the variants are associated. %VAF values as indicators of ctDNA levels. Abbreviations: ND = not detected.

## Data Availability

The data presented in this study are available in anonymized form upon request from the corresponding author. Due to privacy and ethical restrictions, no personally identifiable information can be shared.

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
