# Peer review of "Is Radiographic Aftercare Obsolete? How Testing Positive for ctDNA Can Be a Precedent for Late Relapse, Even in Low-Risk Hormone-Receptor-Positive Breast Cancer"

_ijms, 2025, doi:10.3390/ijms26178498_

Round 1
Reviewer 1 Report
Comments and Suggestions for Authors
1. If this patient has negative PR receptor expression, should this patient be identified as Luminal A? There appears to be divergence in the threshold for determining the classification.
2. The content of the materials and methods is too simple and should be described in further detail.
3. The topic of this article is indeed interesting. Still, the reliability of a single case is insufficient, and there may be individual differences, which is an unavoidable disadvantage of case reports.
none.
Author Response
Reviewer 1
Comments and Suggestions for Authors
- If this patient has negative PR receptor expression, should this patient be identified as Luminal A? There appears to be divergence in the threshold for determining the classification.
Response
We thank the reviewer for his/her careful reading of our manuscript. Indeed, we made an error in the classification of the primary tumor: it is a Luminal B-like carcinoma. Due to the negativity of the progesterone receptor, the carcinoma should be classified within the Luminal B-like cohort (1). We have corrected this accordingly.
- The content of the materials and methods is too simple and should be described in further detail.
Response
Initially, we have limited the materials and methods section to the absolute minimum in order to adhere to the limited word count. We acknowledge, however, that the materials and methods section should include the detailed description of the assay used in this case and have significantly expanded the materials and methods section.
- The topic of this article is indeed interesting. Still, the reliability of a single case is insufficient, and there may be individual differences, which is an unavoidable disadvantage of case reports.
Response:
We thank the reviewer for this important and valid remark. We fully acknowledge the inherent limitations of case reports, including limited generalizability and the potential influence of individual patient factors. However, we believe that this case highlights a clinically relevant and timely aspect of ctDNA-guided surveillance in early breast cancer, which may serve as a basis for future investigations. In our discussion, we explicitly highlight the need for large, prospective trials to evaluate the clinical utility and cost-effectiveness of such approaches. We have also revised the manuscript to more clearly address the exploratory nature of our observations.
- Goldhirsch A, Winer EP, Coates AS, Gelber RD, Piccart-Gebhart M, Thürlimann B, et al. Personalizing the treatment of women with early breast cancer: highlights of the St Gallen International Expert Consensus on the Primary Therapy of Early Breast Cancer 2013. Annals of Oncology. 2013;24(9):2206-23.
Reviewer 2 Report
Comments and Suggestions for Authors
This case report contributes valuable longitudinal data suggesting that circulating tumor DNA (ctDNA) can precede radiographic recurrence in low-risk hormone receptor–positive breast cancer by several years. While the observation is compelling, key methodological details and interpretations remain underexplained. A more rigorous presentation of ctDNA dynamics, variant interpretation, and clinical decision rationale would strengthen the manuscript.
1. How were variants selected for the RaDaR panel, and were any filtered out post hoc? The manuscript states that 53 somatic variants were used for personalized monitoring. Were all detected variants included in the assay, or were certain ones prioritized based on VAF, clonal status, or function? If filtering was done after initial detection, how was overfitting avoided?
2. Was the ctDNA threshold for positivity predefined or empirically derived? The case hinges on persistent ctDNA positivity over five years, yet the manuscript does not state what defined “positive.” Was a fixed %VAF threshold used across all time points? If the threshold was adjusted post hoc, please clarify to what extent this influenced interpretation.
3. What was the clinical trigger for initiating imaging in May 2021 after several years of ctDNA positivity? The manuscript implies that routine imaging was resumed based on retrospective ctDNA detection. Was this decision pre-specified in the study protocol, or reactive to post hoc ctDNA data? The timeline of imaging vs. ctDNA testing must be clarified.
4. How was the 29-fold increase in ctDNA VAF between 05/2021 and 07/2022 calculated? Was this based on a single variant, mean across all detected variants, or a composite eVAF? The claim underpins the argument for "timing" of relapse, so the computational basis must be transparent.
5. Are TTN-AS1 and IGSF1 variants functionally relevant, or only passenger mutations? The two genes are mentioned as "associated with metastasis," but no functional evidence is shown. Are these variants known drivers, or were they selected purely based on detectability? Please provide database links (e.g., COSMIC, OncoKB) or references to justify their mechanistic implication in metastatic recurrence.
Author Response
Reviewer 2
Comments and Suggestions for Authors
This case report contributes valuable longitudinal data suggesting that circulating tumor DNA (ctDNA) can precede radiographic recurrence in low-risk hormone receptor–positive breast cancer by several years. While the observation is compelling, key methodological details and interpretations remain underexplained. A more rigorous presentation of ctDNA dynamics, variant interpretation, and clinical decision rationale would strengthen the manuscript.
- How were variants selected for the RaDaR panel, and were any filtered out post hoc? The manuscript states that 53 somatic variants were used for personalized monitoring. Were all detected variants included in the assay, or were certain ones prioritized based on VAF, clonal status, or function? If filtering was done after initial detection, how was overfitting avoided?
Response
We would like to thank Reviewer 2 for giving us the chance to explain the process of variant selection for designing personalised RaDaR assays. Following WES, raw WES data are converted to FASTQ files followed by read alignment to the human genome (hg38) using a custom analytical pipeline, duplicate marking and somatic variant calling with a proprietary algorithm. Custom filters based on information from publicly available single nucleotide polymorphism (SNP) databases are used to remove germline variants. Subsequently, the proprietary algorithm is used to prioritise tumour-specific somatic variants and select, as part of designing the personalised assay, up to 48 amplicons each targeting at least one such somatic variant. Criteria for variant prioritization and selection include the read depth and allele frequency of each variant as well as assembling a variant set best suited for multiplex PCR to ensure optimal amplification of target regions and subsequent sensitive detection in plasma ctDNA.
For the case discussed in the report, a personalised assay was designed to capture a total of 53 variants. Prior to plasma testing, the designed assay was subjected to quality control (panel qualification; QC) utilizing the same tumour DNA used in WES as well as control genomic DNA. Only variants validated during this QC step (present in tumour, absent in control DNA) were taken forward for plasma testing.
We have significantly expanded the materials and methods section in order to better explain the assay.
- Was the ctDNA threshold for positivity predefined or empirically derived? The case hinges on persistent ctDNA positivity over five years, yet the manuscript does not state what defined “positive.” Was a fixed %VAF threshold used across all time points? If the threshold was adjusted post hoc, please clarify to what extent this influenced interpretation.
Response
Again, we would like to thank Reviewer 2 for this important comment. As explained in the previous comment, plasma calling is based on variants remaining after filtering out those that fail to validate in a patient’s tumour DNA during panel qualification as well as those detected in the matched control DNA. This process not only eliminates the possibility of false positive calls but also allows for the accurate assessment of each sample’s limit of detection through the exact knowledge of the number of variants remaining for use in plasma calling.
In order to call a sample positive or negative for residual disease, RaDaR utilizes a statistical framework that assesses the background noise of each of the remaining variants to determine the statistical significance of the observed mutant counts and their contribution to the calling process. This information is subsequently combined across the entire set of variants validated during panel qualification to call a sample positive when the cumulative statistical score from the contributing variants is above the pre-set threshold defined during RaDaR’s analytical validation. The approach estimates the tumour fraction which is then reported as “% estimated variant allele frequency (%eVAF)”.
The process of calling a patient’s sample positive or negative for residual disease has been previously described. We have now added in the revised manuscript an explanation addressing this comment and provide the following references:
- van Dorp J, Pipinikas C, Suelmann BBM, Mehra N, van Dijk N, Marsico G, et al. High- or low-dose preoperative ipilimumab plus nivolumab in stage III urothelial cancer: the phase 1B NABUCCO trial. Nat Med. 2023;29:588–92.
- Flach S, Howarth K, Hackinger S, Pipinikas C, Ellis P, McLay K, et al. Liquid BIOpsy for MiNimal RESidual DiSease Detection in Head and Neck Squamous Cell Carcinoma (LIONESS)-a personalised circulating tumour DNA analysis in head and neck squamous cell carcinoma. Br J Cancer. 2022;126:1186–95.
- Lipsyc-Sharf M, de Bruin EC, Santos K, McEwen R, Stetson D, Patel A, et al. Circulating Tumor DNA and Late Recurrence in High-Risk Hormone Receptor-Positive, Human Epidermal Growth Factor Receptor 2-Negative Breast Cancer. J Clin Oncol Off J Am Soc Clin Oncol. 2022;40:2408–19.
- What was the clinical trigger for initiating imaging in May 2021 after several years of ctDNA positivity? The manuscript implies that routine imaging was resumed based on retrospective ctDNA detection. Was this decision pre-specified in the study protocol, or reactive to post hoc ctDNA data? The timeline of imaging vs. ctDNA testing must be clarified.
Response:
We thank the reviewer for this very relevant question. The ctDNA analyses in this case report was conducted as part of the SURVIVE pilot study which was exploratory in nature. As the retrospective analyses were carried out in 2021, nearly four years after the initial diagnosis, the study plan included ctDNA assessment and correlation with clinical data. Upon detecting persistent ctDNA positivity in the absence of clinical signs of relapse in 2021, an ethics committee was consulted regarding the appropriate course of action. It was concluded that the findings should be communicated to the patient and that she should be involved in the decision-making process regarding imaging. The patient wished to proceed with staging examinations, which were subsequently initiated in 2021.
We agree that the rationale behind the indications for staging examinations can be explained in more detail and we have revised the manuscript accordingly.
- How was the 29-fold increase in ctDNA VAF between 05/2021 and 07/2022 calculated? Was this based on a single variant, mean across all detected variants, or a composite eVAF? The claim underpins the argument for "timing" of relapse, so the computational basis must be transparent.
Response
The 29-fold increase is based on the calculated eVAF values (May 2021: 0.0044; July 2022: 0.1277; see Supplementary Data). By illustrating the magnitude of this increase, we aim to emphasize the potential clinical relevance of post-adjuvant therapeutic interventions—not only when ctDNA is considered in binary terms (positive/negative), but also and especially when quantitative changes over time are taken into account. We therefore propose to retain this sentence as it underscores the importance of dynamic ctDNA monitoring. However, if the reviewer objects, we are open to removing it accordingly.
- Are TTN-AS1 and IGSF1 variants functionally relevant, or only passenger mutations? The two genes are mentioned as "associated with metastasis," but no functional evidence is shown. Are these variants known drivers, or were they selected purely based on detectability? Please provide database links (e.g., COSMIC, OncoKB) or references to justify their mechanistic implication in metastatic recurrence.
Response
We thank Reviewer 2 for the opportunity to elaborate on the assay used and the selection of the mutations mentioned. The SNVs presented were part of the MRD assay with the aim of maximizing the sensitivity of ctDNA/MRD detection. The goal was not to investigate gene evolution or mutational signatures during carcinogenesis. However, analyzing the origin of the SNVs—i.e., the genes in which they occur—may still offer additional insights into the metastatic process.
According to the COSMIC database, missense mutations of IGSF1 in lobular invasive carcinoma of the breast were observed in 1.79% of tested samples (1/56), and 47.5% of all IGSF1 mutations are classified as missense.
TTN mutations are found in 17.24% of lobular breast cancer samples (10/58), with 70% being missense mutations.
Both SNVs have been identified in the primary tumor and were then detected during the follow-up in the liquid biopsy. This study does not allow mechanistical evidence of both gene products to be relevant for the metastatic process. However, it highlights the potential of ctDNA in the adjuvant setting going beyond the purely binary assessment (detected / not detected) but may help guide surveillance strategies.
Reviewer 3 Report
Comments and Suggestions for Authors
A single patient case report documenting detectable cfDNA using the RaDaR assay following local treatment with a lead-time detection almost 5 years before actual clinical relapse.
- There is no discussion of the number needed to test vs those that are positive for ctDNA given a Luminal B, stage 2 situation. Most trials testing the concept have stopped testing stage 2, because the positive rate is so low. They have focused on stage 3 or high nodal stage 2.
- Line 51 states the tumor was Luminal A despite a PR of 0%. Later in the paper, the same characteristics are described as Luminal B (line 81). What is the basis for stating either?
- was there consideration for XRT given the regional relapse without metastatic disease?
- Please describe the strengths and weaknesses of the RaDaR test vs other assays used to detect ctDNA. For example, this assay does not evaluate the development of ESR1 mutations, which might affect the choice of fulvestrant as treatment.
- The use of actual dates for scans and assays might violate HIPAA rules (in the US). Perhaps better to delineate the year of diagnosis and then relative months of f/u.
Author Response
Reviewer 3
Comments and Suggestions for Authors
A single patient case report documenting detectable cfDNA using the RaDaR assay following local treatment with a lead-time detection almost 5 years before actual clinical relapse.
- There is no discussion of the number needed to test vs those that are positive for ctDNA given a Luminal B, stage 2 situation. Most trials testing the concept have stopped testing stage 2, because the positive rate is so low. They have focused on stage 3 or high nodal stage 2.
Response
We thank Reviewer 3 for this highly relevant and insightful comment. We are indeed aware of the low ctDNA positivity rates in the adjuvant setting of low-risk breast cancer, which have posed significant challenges for previous studies and continue to do so for ongoing trials (e.g., the SURVIVE study, NCT05658172).
The aim of our case report, however, is not to provide evidence for broad applicability, but rather to illustrate that ctDNA-guided surveillance may offer a potential clinical window of opportunity even in early-stage and low-risk breast cancer patients. We acknowledge that this study does not allow for an assessment of the number needed to treat (NNT) or the economic implications of a widespread ctDNA-based surveillance strategy.
To reflect this important limitation, we have added a clarifying statement to the discussion section.
- Line 51 states the tumour was Luminal A despite a PR of 0%. Later in the paper, the same characteristics are described as Luminal B (line 81). What is the basis for stating either?
Response:
As stated above (Response to Reviewer 1), we have made an error in the characterisation of the primary tumor as Luminal A, based on the St Gallen Consensus of 2013 (1). We have revised the manuscript accordingly.
- was there consideration for XRT given the regional relapse without metastatic disease?
Response:
We thank Reviewer 3 for giving us the opportunity to explain the rationale behind the treatment decision after the detection of locoregional relapse.
The systemic therapy of letrozole + abemaciclib was given after careful consideration and discussion with the patient. As positive level III axillary lymph nodes are considered locally advanced disease, the decision for systemic therapy is in accordance with the approved indication.
The decision for / against surgery is in our view the subject of ongoing clinical debate in the emerging field of ctDNA positivity and MRD. The prolonged ctDNA positivity reflects a systemic course of the disease. A surgical approach aiming for R0 resection would not allow for in-label adjuvant administration of an estrogen receptor degrader such as Fulvestrant in combination with abemaciclib, thus maybe leading to early second relapse, as can be expected in this case of AI resistance.
Future investigations will determine whether patients with locoregional relapse detected through ctDNA positivity exhibit different clinical courses compared to those with conventionally detected relapses, and whether treatment strategies should be adapted accordingly.
- Please describe the strengths and weaknesses of the RaDaR test vs other assays used to detect ctDNA. For example, this assay does not evaluate the development of ESR1 mutations, which might affect the choice of fulvestrant as treatment.
Response
Although we feel a direct comparison of the RaDaR assay with other MRD assays is out of the scope of this case report study, we would like to thank Reviewer 3 for this comment and for giving us the opportunity to describe the RaDaR assay and explain how the personalised assays are designed.
RaDaR® is a multiplex PCR and ultra-deep next-generation sequencing (NGS) liquid biopsy assay specifically designed to detect molecular residual disease and monitor for disease recurrence,
The assay employs a tumour-informed approach, meaning that a tumour specimen from a patient is subjected to whole exome sequencing to identify tumour-specific somatic variants and allow the design of personalised assays. The process of designing these assays utilises a proprietary algorithm for somatic variant calling. Identified somatic variants are then prioritised and selected based on certain criteria including the read depth and allele frequency of each variant as well as assembling a variant set best suited for multiplex PCR to ensure optimal amplification of all target regions. This approach, as it has been demonstrated through the assay’s analytical development and validation, can achieve improved sensitivity at sample level for ctDNA detection down to 11 parts per million (ppm, 0.0011% variant allele frequency). Due to the aim of the assay and the selection criteria considered when designing a personalised panel, variants in genes, such as ESR1 which are known to be associated with treatment resistance, as well as variants in genes with available targetable therapies might not be part of the final panel variant set used in plasma ctDNA calling.
We have now added in the methods a description on the process of variant selection for panel design which we hope addresses in full Reviewer’s 3 comment.
- The use of actual dates for scans and assays might violate HIPAA rules (in the US). Perhaps better to delineate the year of diagnosis and then relative months of f/u.
Response
We thank Reviewer 3 for the thoughtful comment. We have accordingly rephrased the time points.
Round 2
Reviewer 3 Report
Comments and Suggestions for Authors
thank you for addressing most of my concerns. The case report is of interest, although I am concerned regarding the overuse of these assays as a free-for-all outside of a clinical trial. It is critical that ctDNA be rigorously tested for its clinical utility and not merely for its accuracy.